# Genetically Related *Mycobacterium bovis* Strains Displayed Differential Intracellular Growth in Bovine Macrophages

**DOI:** 10.3390/vetsci6040081

**Published:** 2019-10-18

**Authors:** Alejandro Benítez-Guzmán, Hugo Esquivel-Solís, Cecilia Romero-Torres, Camila Arriaga-Díaz, José A. Gutiérrez-Pabello

**Affiliations:** 1Facultad de Medicina Veterinaria y Zootecnia, Universidad Nacional Autónoma de México, Ciudad de México 04510, Mexico; 2Biotecnología Médica y Farmacéutica, Centro de Investigación y Asistencia en Tecnología y Diseño del Estado de Jalisco (CIATEJ), Guadalajara 44270, Mexico; 3Instituto Nacional de Investigaciones Forestales Agrícolas y Pecuarias, CENID Microbiología, Ciudad de México 05110, Mexico

**Keywords:** *M. bovis*, spoligotype, virulence, intracellular replication, macrophages

## Abstract

Molecular typing of bacterial isolates provides a powerful approach for distinguishing *Mycobacterium bovis (M. bovis)* genotypes. It is known that *M. bovis* strain virulence plays a role in prevalence and spread of the disease, suggesting that strain virulence and prevailing genotypes are associated. However, it is not well understood whether strain virulence correlates with particular genotypes. In this study, we assessed the in vitro intracellular growth of 18 *M. bovis* isolates in bovine macrophages as an indicator of bacterial virulence and sought a relationship with the genotype identified by spoligotyping. We found 14 different spoligotypes—11 were already known and three spoligotypes had never been reported before. We identified 2 clusters that were phylogenetically related, containing 10 and 6 strains, respectively, and 2 orphan strains. Intracellular growth and phagocytic rates of 18 *M. bovis* strains were heterogeneous. Our results suggest that *M. bovis* intracellular growth and phagocytosis are independent of the bacterial lineage identified by spoligotyping.

## 1. Introduction

Bovine tuberculosis (BTB) is an infectious disease originated by the presence of *Mycobacterium bovis* in cattle. BTB generates high economic losses in livestock and affects animal health policies. This entity has spread worldwide, within a broad range of hosts, including humans, livestock, and wildlife [1]. According to the available disease timelines in the http.mbovis.org information database, *M. bovis* has been isolated from countries in all continents, except Antarctica [2].

*M. bovis* strains differ depending on the host they are infecting and the geographic region they are located in. Molecular typing of bacterial isolates based on polymorphisms in genomic DNA (genotyping) provides a powerful approach in distinguishing *M. bovis s*trains, and may further yet provide valuable insights into the maintenance and transmission of infection [3]. The most common epidemiological molecular typing methods for members of the *Mycobacterium tuberculosis* complex are: (a) insertion sequence 6110-based restriction fragment length polymorphism (IS6110-RFLP), (b) spacer oligonucleotide typing (spoligotyping), and (c) the analysis of the copy number of mycobacterial interspersed repetitive unit-variable number tandem repeats (MIRU-VNTRs). Spoligotyping is the most commonly used tool for *M. bovis* genotyping because it is easy to perform and has the capability to discriminate between strains, since it excludes other species from the *M. tuberculosis* complex [4,5].

*Mycobacterium tuberculosis* strains differ in the way they invade, multiply, and persist in the host’s tissues. For example, the Euro-American lineage of *Mycobacterium tuberculosis* is less capable of producing meningeal tuberculosis than those strains predominantly originating in Asia [6]. The Beijing strains grow significantly faster in human macrophages and have a higher number of colony-forming units than those of non-Beijing strains, and are correlated with more severe lung lesions in mouse infections [7,8]. When we combine all of this data, it is suggestive of an association between specific genotypes and virulence.

The macrophages derived from peripheral blood have been use as a key of innate immunity to tuberculosis and a phenotypic marker of cattle susceptibility to *M. bovis* infection. Macrophages from different animals had differential intracellular control of *M. bovis* strains, depending on virulence. Our previous study showed differences in the bacterial survival rate of 65% and 100% for the avirulent *M. bovis* BCG and the virulent field isolate known as “El Paso”, respectively [9,10]. Macrophages from animals that better control bacterial replication were found to produce a higher oxidative burst, having greater bacteriostatic activity, producing higher concentrations of nitric oxide, and having a higher pro-inflammatory cytokine gene expression [11,12,13]. Results from our laboratory demonstrated that the *M. bovis* strain virulence classification obtained by a well-characterized mouse model of progressive pulmonary tuberculosis [14] was also replicable in the bovine macrophage in vitro model. Virulent strains had higher intracellular growth than the attenuated ones [15]. Therefore, bacterial intracellular survival and replication in an in vitro macrophage infection assay may be considered as a correlate of virulence [8,16,17].

According to literature, some *M. bovis* genotypes are more predominant in certain geographical regions than others within a cattle population [18]. We hypothesize that virulence influences the persistence and prevalence of strains, and that virulence is related to lineage. However, it is not well understood whether strain virulence correlates directly with specific genotypes. In this study, we aimed to assess intracellular growth of *M. bovis* isolates in macrophages of naturally resistant cattle as an indicator of bacterial virulence, and evaluate the possible correlation with the strain genotype identified by spoligotyping. Our results demonstrated that strains belonging to the same phylogenetically related clusters showed a differential intracellular survival/replication rate, suggesting that spacer composition does not influence *M. bovis* intracellular growth in bovine macrophages.

## 2. Methods

### 2.1. Ethic Statement

All animal procedures were performed according to the Facultad de Medicina Veterinaria y Zootecnia from the Universidad Nacional Autónoma de México (FMVZ-UNAM) board’s statements on animal research, based on the Mexican law on animal studies. Ethical approval for this study was obtained from the FMVZ-UNAM Institutional Committee for Care and Use of Experimental Animals (CICUAE) (JAGP-2002).

### 2.2. Macrophage Culture

Venous peripheral blood was obtained from healthy adult cattle, from a tuberculosis-free herd housed at the facilities of the Research and Teaching Center (CEPIPSA) of the Universidad Nacional Autónoma de México (UNAM). Macrophages were obtained from peripheral blood mononuclear cells (PBMC), as described before [11]. Blood was collected from the jugular vein into 60 mL syringes containing an acid-citrate-dextrose solution and was centrifuged at 1000 ×*g* for 30 min. Buffy coats were diluted in 30 mL of citrated PBS, layered onto 15 mL of Percoll (Pharmacia, Uppsala, Sweden) at a specific density of 1.077, and centrifuged at 1200 ×*g* for 25 min. PBMC were then removed from the interface between the plasma and Percoll solution, pooled, diluted in 50 mL of citrated phosphate buffered saline (PBS), and centrifuged at 500 ×*g* for 15 min. The cell pellets were then washed three times with citrated PBS at 500 ×*g* for 10 min, suspended in Roswell Park Memorial Institute (RPMI) (Gibco, New York, NY, USA) supplemented with 5 mM L-glutamine (Gibco, New York, NY, USA), 5 mM non-essential amino acids, and 5 mM sodium pyruvate (Gibco, New York, NY, USA) (CRPMI) containing 4% autologous serum to facilitate adherence, and cultured overnight at 37 °C and 5% CO_2_. Non-adherent cells were then removed by three washes with prewarmed PBS, and adherent monocytes were cultured, as described previously, in CRPMI plus 12% autologous serum for 12 days until they differentiated to macrophages. Flasks were chilled on ice for 45 min and macrophages were harvested by repeated gentle pipetting.

### 2.3. Mycobacterium Bovis Strains

The strains used in this study are listed in Table 1. Strains were obtained by convenience sampling performed by official veterinarians in the field. All strains were isolated from visible lesions observed at the slaughterhouse. Strains from the same geographical region came from different farms. Strains were grown in Middlebrook 7H11 medium (Difco Laboratories, Detroit, MI, USA) with oleica acid, dextrose and catalase (OADC) (Becton and Dickinson, Sparks, NV, USA) from 18 to 21 days and then were seeded for 19 days in 7H9 broth supplemented with 0.05% Tween 80 and OADC (Becton and Dickinson, Sparks, NV, USA). Bacteria were suspended in CRPMI, passed twice through a 27-gauge needle, and sonicated for 30 s in order to disrupt the clumps. Aliquots of 1 mL were stored at −80 °C. Inoculums were titrated by plating colony-forming unit (CFU) serial dilutions on Middlebrook 7H11 (Difco Laboratories, Detroit, MI, USA) plus 10% of OADC.

### 2.4. Spoligotyping

DNA was obtained by proteinase K-guanidine hydrochloride method. Spoligotyping was performed following a standard protocol (Kamerbeek et al., 1997). Briefly, bacilli DNA was amplified with AmpliTaq DNA polymerase (PerkinElmer, Norwalk, CT, USA) in a 50 μL PCR mix containing 5 μL of 10 × reaction buffer (100 μM Tris–HCl, pH 8.3, 500 M KCl, 15 μM MgCl_2_, 0.01% (w/v) gelatin, 0.2 μM of dNTP, 20 pmol each primer DRa (5′ -GGT TTT GGG TCT GAC GAC- 3′) marked with biotin at the 5′ DRb (5′-CCG AGA GGG GAC GGA AAC-3′), and 10 ng of mycobacterial DNA. The mixture was heated in a Gene Amp PCR system 2400 (PerkinElmer, Norwalk, CT, USA) at 96 °C for 3 min and subjected to 30 cycles at 96 °C for 1 min, 55 °C for 1 min, and 72 °C for 40 s, until reaching 72 °C for 10 min. The amplified DNA was visualized after electrophoresis in a Sodium Dodecyl sulfate ( SDS) polyacrylamide 12% gel stained with silver nitrate at 120 V for 90 min. Twenty microliters of PCR product was added to 150 μL of 2 × SSPE-0.1% SDS (15 mM sodium chloride, 1 mM sodium phosphate, 1mM EDTA, 0.1% SDS) and heat denatured at 99 °C for 10 min, and applied to a nylon membrane to which 37 spacer sequences from *M. tuberculosis* H37Rv and 6 spacer sequences from *M. bovis* BCG were covalently bound (ISOGEN, Maarssen, The Netherlands). This membrane was then placed in a miniblotter MN45 (Immunetics; Cambridge, MA, USA) in such a way that the slots were perpendicular to the line pattern of the previously applied oligonucleotides. The membranes were then incubated in 10 mL of 2 × SSPE-0.5% SDS plus 5 μL of streptavidin-peroxidase conjugate for 45 min at 42 °C. For detection of hybridizing DNA, enhanced chemiluminescence (ECL) detection liquid (Amersham Biosciences; Piscataway, NJ, USA) was used, followed by exposure to X-ray film (Kodak, Rochester, NY, USA) for 12 min.

### 2.5. Bactericidal Assay

The bactericidal assay was performed as described previously [11]. Briefly, macrophage monolayers in 60-microwell tissue culture plates (Nunc, Roskilde, Denmark) were infected with each strain of *M. bovis* at a multiplicity of infection (MOI) of 10:1, centrifuged at 200 ×*g* for 10 min, and incubated at 37 °C with 95% humidity and 5% CO_2_ for 4 h (the time previously determined to allow macrophage phagocytosis of mycobacteria). After this time (considered 0 h post-infection), cells were washed 5 times to remove extracellular bacteria with 5 μL of fresh CRPMI and incubated for 24 h at 37 °C. To calculate mycobacteria intracellular growth, the number of intracellular CFU at the end of the assay (24 h) was divided by the total number of CFU at the beginning (0 h) and expressed as percentage. CFU were recovered, serial diluted, plated onto Middlebrook 7H11 plus 10% OADC agar, and cultured for 21 days for counting after lysis of macrophages with 0.5% Tween 20 at 0 h and 24 h post-infection. We also estimated the average number of mycobacteria per macrophage after phagocytosis by dividing the total number of CFU at 0 h by the total number of seeded macrophages. The results are the average of three independent experiments, each one with three internal replicates.

### 2.6. Statistical Analysis

A data analysis composed of matrix 1 or matrix 0 was built on the bases of presence (1) or absence (0), respectively, of fragments hybridized in the spoligotyping. Unweighted pairwise group method (UPGMA) analysis was performed to build the dendrogram with multivariate statistical analysis, using squared Euclidean distance with the hierarchical cluster option and nearest neighbor.

Comparisons within rates of phagocytosis and intracellular growth of all *M. bovis* strains were performed using one-way ANOVA tests, as well as Tukey’s post hoc test for multiple comparisons between all strains and within spolygotype clusters. Fisher’s least significant difference test was done for comparisons of each strain rate of intracellular growth versus 100%, the cutoff reference value that signifies a strain’s replication into the macrophage. A *p* value of 0.05 or less was considered statistically significant.

## 3. Results

### 3.1. Spoligotyping Identified Strains Already Present in the Region and Three New Spoligotypes

We analyzed 18 different *M. bovis* strains isolated from tuberculous lesions of dairy cattle and found 14 different spoligotypes, among them were three spoligotypes that had never been before reported. Spoligotyping results are summarized in Table 1. *M. bovis* spoligotypes identified in this study correspond to spoligotypes reported previously in Mexico. Pattern information of the new strains was submitted to the *M. bovis* database strains 9524 (SB2353), QP129, 9930, 9926 (SB2352), and 9918 (SB2351). Several of the identified spoligotypes had been described previously in other countries, such as Argentina (SB0145 and SB0140) and the United Kingdom (SB0669, SB0673, and SB0140). The UPGMA agglomerative hierarchical clustering algorithm analysis identified 2 defined clusters, showing a dissimilarity ratio (0–5). Cluster one comprises 10 strains, ordered according to their kin relationship by the spoligopattern (T3, T41C, 9920, CO3, CO5, 9912, 9917, 9563, 9914, 9918). The second cluster is composed of 6 strains, in order of their relationship (9926, QP129, 9930, 9524, 9916, 9922). Two *M. bovis* strains (9927 and 163QR) are not related to any others; thus, they do not belong to any cluster (Figure 1).

### 3.2. Mycobacterium Bovis Strains Showed Dissimilar Phagocytosis and Intracellular Survival/Replication Rates

Macrophage phagocytic rates of *M. bovis* strains showed a dissimilar pattern. They were grouped into five groups based on the statistical differences shown in the ANOVA test (Figure 2, Table 2). Phagocytic rates were as follows: the first group consists of 4 strains with less than one mycobacteria per macrophage (0.16 to 0.87); the second group comprises 6 strains with more than one and less than 2 mycobacteria per macrophage (1.2 to 1.76); the third group consists of 2 strains with 2.21 and 2.53; the fourth group includes 5 strains with more than 3 and less than 4 mycobacteria per macrophage (3.22 to 3.68); and the fifth group includes one strain with 7.43 mycobacteria per macrophage (Figure 2). Based on the intracellular survival/replication rate, strains were grouped in three blocks according to the statistical differences shown by the ANOVA test. The first group consists of one strain (1/18) with less than 35.6% intracellular survival/replication; interestingly enough, this strain has an intracellular survival/replication pattern similar to the avirulent *M. bovis* BCG. The second group is the largest in number, comprising 14/18 strains, with intracellular survival/replication between 53% to 129.3%. The last group had 3/18 strains, and had an intracellular survival/replication rate between 146% and 158.6% (Figure 3). We defined the analysis by spoligotype clusters in order to seek a relation between the genotype and their capability of surviving inside macrophages (suggesting virulence), but were unable to identify such a relation. Strains belonging to the same spoligotype cluster had different phagocytic and intracellular survival/replication rates between them (*p* ≤ 0.05).

### 3.3. Spacer Composition Does Not Influence M. bovis Intracellular Survival/Replication in Bovine Macrophages

Strains belonging to the same spoligotype cluster and even to the same spoligotype have different intracellular survival/replication rates. Statistical analysis did not show a specific pattern of association for intracellular survival/replication and phagocytosis with bacterial spoligotype identity. Our results suggest that *M. bovis* intracellular survival/replication and phagocytosis are independent of the bacterial spoligotype.

## 4. Discussion

Most of the molecular tools used to classify the *Mycobacterium tuberculosis* complex isolates rely on sequences that, per se, do not code for virulence factors. For instance, spoligotyping is based on identification of the direct repeat (DR) locus. The DR region is composed of multiple direct variant repeats (DVRs), each of which is composed of a 36-bp direct repeat (DR) plus a non-repetitive spacer sequence of a similar size. Maintenance of this region in the *M. tuberculosis* complex and the strong sequence conservation of the DRs and spacers among clinical isolates are features that suggest a biological function of this region [19,20,21]. At the present time, the function of the DR region is not known; however, it may play a role in regulating gene expression, and therefore influence, to some extent, the bacteria’s ability to survive the killing machine that is the macrophage.

Our results identified two spoligotype clusters. Strains belonging to each cluster have a small variation in their spacer composition; in some cases, only one spacer is missing, suggesting that the influence of the DR locus in bacterial performance may be similar. In the present study, we used macrophages from a donor with a phenotype of natural disease resistance to intracellular bacterial pathogens, in order to evaluate intracellular survival/replication as a means of identifying virulence of *M. bovis* strains. Previous results from our laboratory linked the resistance phenotype with some immunological markers, such as nitric oxide. Our experiments not only explore the nitric oxide production on resting macrophages, but also on classical and alternative activated macrophages. Macrophages from resistant donors produced more nitric oxide than susceptible cells. Our previous study demonstrated that nitric oxide is a major determinant of bacterial survival/replication in macrophages [11], [12].

According to our results, phagocytic efficiency and intracellular replication are independent variables within the spoligotype clusters, while strains with similar phagocytic index had different intracellular survival/replication rates. Higher intracellular survival/replication indicates that bacteria possess mechanisms to avoid the macrophage killing machinery; therefore, it may be postulated that the bacteria harbor higher virulence. Our working hypothesis proposed that strains with similar intracellular survival/replication may belong to the same spoligotype cluster; however, our observations did not find any evidence that connected both features. Moreover, strains in the same cluster and even with the same spoligotype showed a differential intracellular survival/replication, suggesting that spoligotype spacer composition could not be associated with the strain’s ability to survive in macrophages. Similar results from a field study support our observations, suggesting that virulence is divergent among strains belonging to the same spoligotype. A macroscopical lesion score showed a variation from 3 to 43 in the degree of tissue damage induced by more than 60 strains isolated from cattle that correspond to spoligotype SB0140 [18,22]. The presence of strains belonging to a particular spoligotype in cattle populations may indicate that bacteria possess a basic group of virulence genes that they can pass among different individuals; however, it is not indicative of the full repertoire of virulence factors that each strain possess. In addition, there is variation in the genetic composition of the host that is not contemplated in field studies. In our in vitro study, we evaluated intracellular survival/replication using macrophages from the same donor, providing us a more comparable setting. A possible pitfall in our study is the limited number of strains included, however, studies with a higher number of strains also observed similar results [18]. In a closely related study, 3 strains from the Beijing family with a dissimilar IS6110 pattern had a similar infective performance when they were inoculated in mice. To summarize, these data indicate that genotype patterns do not correlate with bacterial virulence performance. Further analysis, such as next-generation sequencing, indicates strong association between genotype and virulence [23]. *M. bovis* spoligotypes identified in this study correspond to spoligotypes reported previously in Mexico, with the exception of SB1498. We also found 3 spoligotype patterns that had not been reported before. All of the spoligotypes lack spacer number 11, suggesting that those strains are part of the European 1 clonal complex (EU1), which is believed to have its origin in the United Kingdom. In addition, it has been hypothesized that this group of strains is derived from the SB0140 spoligotype [24,25,26]. Strains from the same geographical region came from different farms, suggesting that the spoligotype represented is prevalent in the region. However, in some other instances, strains from the same region represent different spoligotypes, suggesting a high spoligotype variation in one geographical area. Our study is by no means an epidemiological survey; however, our results correspond with previous studies that showed high spoligotype diversity among isolates in Mexico [27,28,29,30].

## 5. Conclusions

Our findings showed that spacer composition similarity is not associated with the bacterial ability to survive inside macrophages from a donor with capacities to control intracellular growth of *M. bovis.*

## Figures and Tables

**Figure 1 vetsci-06-00081-f001:**
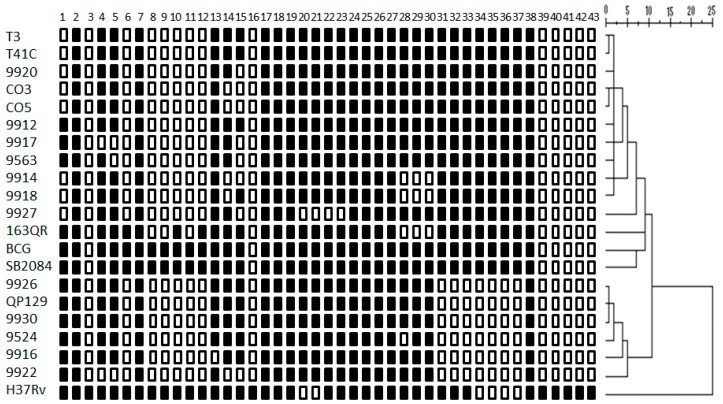
Tree of spoligotype patterns of *M. bovis* strains. The unweighted pairwise group method analysis (UPGMA) was performed with multivariate statistical package software using squared Euclidean distance with the hierarchical cluster option and nearest neighbor. The agglomerative hierarchical clustering algorithm analysis identified 2 defined clusters, showing a dissimilarity ratio (0–3). *M. bovis* field strain SB2084, BCG Montreal strain, and *Mycobacterium tuberculosis* H37Rv international reference strain were used as controls.

**Figure 2 vetsci-06-00081-f002:**
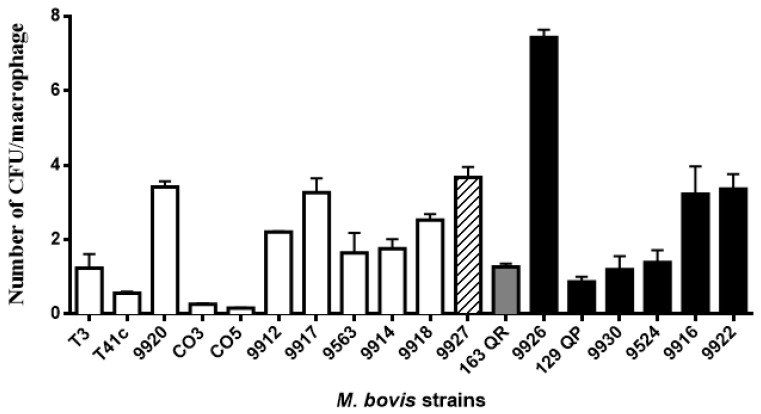
Macrophage phagocytic rates of *M. bovis* strains showed a dissimilar pattern. Macrophage monolayers were infected with *M. bovis* at a multiplicity of infection (MOI) ratio of 10:1 for 4 h. Bacterial phagocytosis was calculated by plating serial dilutions of the live intracellular bacteria released from the macrophages after treatment with 0.5% Tween 20. Total number of colony-forming units was divided by the total number of macrophages to obtain an average bacterial concentration per macrophage. Phagocytic rates were grouped into different blocks based on statistical differences. The color pattern of the bar indicates the group to which it belongs. Results are the average of three independent experiments, each one with three internal replicates.

**Figure 3 vetsci-06-00081-f003:**
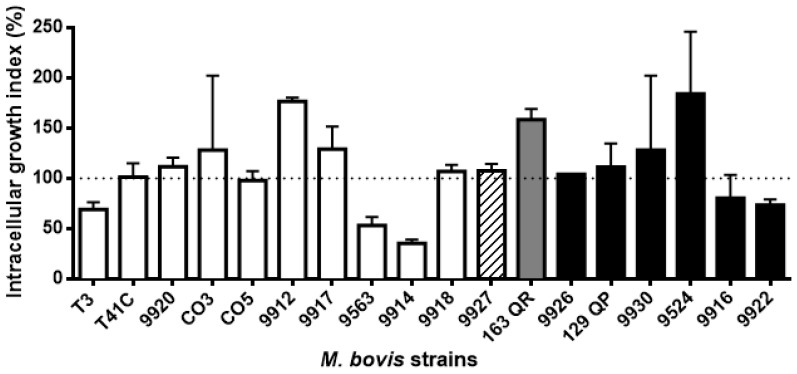
*M. bovis* intracellular survival/replication rates showed a high degree of variation among strains. Macrophage monolayers were infected with *M. bovis* at a multiplicity of infection (MOI) ratio of 10:1 for 4 h; this was considered time 0 h. The cells were harvested at 0 h and 24 h post-infection for analysis. Bacterial growth was calculated as the ratio of the total number of intracellular bacteria at the end of the assay to the total number of bacteria at the start the assay, expressed as percentage. Results are the average of three independent experiments, each one with three internal replicates. Statistical analysis was performed using Tukey’s test. A *p* value of 0.05 or less was considered statistically significant. The color pattern of each bar indicates the group to which it belongs.

**Table 1 vetsci-06-00081-t001:** *M. bovis* strains used in this study. Data summarize spoligotype identity and strain origin. Nil (spoligotype not reported before). Strain control: *M. bovis* field strain SB2084 and Bacillus Calmette-Guerin (BCG) Montreal.

Strain ID	Spoligotype ID	Place of Isolation	Strain Origin *
T3	SB0673	Torreón	United Kingdom
T41C	SB0673	Torreón	United Kingdom
9920	SB0669	Jalisco	United Kingdom
CO3	SB1112	Edo de Mex	Mexico
CO5	SB1112	Edo de Mex	Mexico
9912	SB0140	Nayarit	Argentina/United Kingdom
9917	SB1816	Jalisco	Mexico
9563	SB1498	Jalisco	Mexico
9914	SB1811	Jalisco	Mexico
9918	SB2351 (Nil)	Jalisco	Mexico
9927	SB1115	Jalisco	Mexico
163QR	SB1503	Querétaro	Mexico
BCG	SB0120		
Control	SB2084		Mexico
9926	SB2352 (Nil)	Jalisco	Mexico
QP129	SB2352 (Nil)	Querétaro	Mexico
9930	SB2352 (Nil)	Jalisco	Mexico
9524	SB2353 (Nil)	Edo de Mex	Mexico
9916	SB1819	Jalisco	Mexico
9922	SB1815	Mexico	Mexico

Note: * Country where isolation was first reported.

**Table 2 vetsci-06-00081-t002:** Macrophage phagocytic and intracellular growth rates. Data summarize percentages of bacterial phagocytosis per macrophage and the intracellular replication. In both cases the strains were grouped according to statistical differences using Tukey’s test.

Strain ID	Spoligotype ID	Phagocytosis %	Statistical Group	Intracellular Growth %	Statistical Group
T3	SB0673	1.24	2	69.1	2
T41c	SB0673	0.57	1	101.3	2
9920	SB0669	3.42	4	111.6	2
CO3	SB1112	0.27	1	128.2	2
CO5	SB1112	0.17	1	98.1	2
9912	SB0140	2.21	3	176.5	3
9917	SB1816	3.27	4	129.3	2
9563	SB1498	1.65	2	53.2	2
9914	SB1811	1.76	2	35.6	1
9918	SB2351	2.53	3	107.0	2
9927	SB1115	3.68	4	107.7	2
163QR	SB1503	1.27	2	158.6	3
9926	SB2352	7.43	5	104.0	2
129QP	SB2352	0.87	1	111.4	2
9930	SB2352	1.20	2	128.2	2
9524	SB2353	1.39	2	184.0	3
9916	SB1819	3.22	4	80.6	2
9922	SB1815	3.36	4	73.7	2

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
