# Peer review of "Genetically Related Mycobacterium bovis Strains Displayed Differential Intracellular Growth in Bovine Macrophages"

_vetsci, 2019, doi:10.3390/vetsci6040081_

Round 1

Reviewer 1 Report

The manuscript describes a comparison of 18 strains of M. bovis isolated in Mexico.  The strains were genotyped using spoligotyping and their ability to be phagocytosed by bovine macrophages and survive/proliferate over 24h were compared.  Fourteen spoligotypes were identified, which fell into two major clusters.  However, there was little correlation between spoligotype and either the phagocytic index or intracellular survival/replication.  Furthermore, there was no correlation between phagocytic index and survival/replication.

Although a relatively small study, the subject is of interest to the field of Mycobacterial research.  It is well written and I do not have any major issues with the manuscript.

Minor issues

Line 22 & elsewhere – in vitro and in vivo should be in italics

Line 27 & elsewhere – there is a muddled use of Mycobacterium bovis and M. bovis throughout the manuscript. 

Line 28 – It should also be mentioned that there was no correlation between the initial uptake of M. bovis and their survival/replication after 24h.

Line 52 – clarify what is meant by ‘a higher number of colony-forming units’.  Do you mean that the infected lung contains more bacteria?

Line 75 – why is phylogenetically in italics?

Line 111 – should it be Middlebrook 7H11 agar rather than medium?

Line 119 – how was the mycobacterial gDNA isolated?

Line 159 – clarify and rephrase ‘strain survived to the macrophage attempt to kill it.’

Line 168 – Rephrase to ‘Several of the identified spoligotypes have been described previously in other countries, for example Argentina ……..’

Fig. 2 & 3 – The figures may be easier to interpret if ordered on the x axis according to their ability to be taken up (Fig 2) and intracellular growth (Fig 3).  The colouring of the bars shows which indicate the cluster the strains belonged to. It would remove the requirement for all the additional numbers above the bars, which in the case of Fig. 3 are rather messy. 

Line 193 – 35.6% of intracellular replication suggests 35.6% increase in bacterial numbers.  Change to intracellular survival.

Line 196 – survival rates should be 158.6 to 184%. 

Line 196 - Is survival the appropriate term to use here?  It would appear that replication is occurring.  The number of intracellular M. bovis will depend on the interplay between M. bovis replication and M. bovis killing by the mycobactericidal activity of the macrophage.  This has been demonstrated by comparing the quantification of live bacteria (by CFU) and genome copies (by qPCR) (Jensen et al., 2018 Infect. Immun. 86:e00385-17).  Therefore survival/replication may be a more accurate term to use throughout the manuscript.

Line 224 – for clarity change 'spoligotype group' for 'spoligotype cluster'

Line 226 – change ‘point out’ to ‘suggest’

Line 262 – The authors may not be aware of the work published on M. bovis isolates from Northern Ireland.  In this study isolates of SB0140, further distinguished by VNTR, were shown to exhibit significantly different levels of virulence, measured as the proportion of reactor animals with visible granulomas/lesions (Wright et al., 2013 PLoS One 8:e74503), which provides further evidence that spoligotypes are not a good indicator or virulence. 

Punctuation/Typos

Line 32 – unnecessary comma after tuberculosis

Line 60 – unnecessary full stop after respectively

Line 63 – unnecessary comma after demonstrated

Line 74 – should read ‘….. belonging to the same ….’

Line 75 – unnecessary comma after clusters

Line 167 – extra bracket after database

Line 216 – missing full stop after analysis

Line 226 – unnecessary comma after out

Line 235 – unnecessary comma after isolates

Line 236 – should read ‘At the present time the function …..’

Line 240 – unnecessary comma after cases

Line 240 – should read ‘suggesting that the influence….’

Line 252 – unnecessary comma after indicates

Line 262 – the sentence should start with ‘The’

Line 269 – change close to closely

Line 276 – full stop in the wrong place

Author Response

Response to reviewers (in red) point by point. Please see the attachment with the manuscript

Reviewer 1

Minor issues

Line 22 & elsewhere – in vitro and in vivo should be in italics

Done

Line 27 & elsewhere – there is a muddled use of Mycobacterium bovis and M. bovis throughout the manuscript. 

Done

Line 28 – It should also be mentioned that there was no correlation between the initial uptake of M. bovis and their survival/replication after 24h.

Done

Line 52 – clarify what is meant by ‘a higher number of colony-forming units’.  Do you mean that the infected lung contains more bacteria?

Done

Line 75 – why is phylogenetically in italics?

Was an error

Line 111 – should it be Middlebrook 7H11 agar rather than medium?

Done

Line 119 – how was the mycobacterial gDNA isolated?

The method was specified

Line 159 – clarify and rephrase ‘strain survived to the macrophage attempt to kill it.’

Done

Line 168 – Rephrase to ‘Several of the identified spoligotypes have been described previously in other countries, for example Argentina ……..’

Done

Fig. 2 & 3 – The figures may be easier to interpret if ordered on the x axis according to their ability to be taken up (Fig 2) and intracellular growth (Fig 3).  The colouring of the bars shows which indicate the cluster the strains belonged to. It would remove the requirement for all the additional numbers above the bars, which in the case of Fig. 3 are rather messy. 

After modifications suggest by the reviewer, the numbers above de bars was deleted, however the format of the graphs it was preserved, because we found no advantages in the interpretation.

Line 193 – 35.6% of intracellular replication suggests 35.6% increase in bacterial numbers.  Change to intracellular survival.

Done

Line 196 – survival rates should be 158.6 to 184%. 

Done

Line 196 - Is survival the appropriate term to use here?  It would appear that replication is occurring.  The number of intracellular M. bovis will depend on the interplay between M. bovis replication and M. bovis killing by the mycobactericidal activity of the macrophage.  This has been demonstrated by comparing the quantification of live bacteria (by CFU) and genome copies (by qPCR) (Jensen et al., 2018 Infect. Immun. 86:e00385-17).  Therefore survival/replication may be a more accurate term to use throughout the manuscript.

Done

Line 224 – for clarity change 'spoligotype group' for 'spoligotype cluster'

Done

Line 226 – change ‘point out’ to ‘suggest’

Done

Line 262 – The authors may not be aware of the work published on M. bovis isolates from Northern Ireland.  In this study isolates of SB0140, further distinguished by VNTR, were shown to exhibit significantly different levels of virulence, measured as the proportion of reactor animals with visible granulomas/lesions (Wright et al., 2013 PLoS One 8:e74503), which provides further evidence that spoligotypes are not a good indicator or virulence.

Done 

Punctuation/Typos

Line 32 – unnecessary comma after tuberculosis

Done

Line 60 – unnecessary full stop after respectively

Done

Line 63 – unnecessary comma after demonstrated

Done

Line 74 – should read ‘….. belonging to the same ….’

Done

Line 75 – unnecessary comma after clusters

Done

Line 167 – extra bracket after database

Delete

Line 216 – missing full stop after analysis

Done

Line 226 – unnecessary comma after out

Delete

Line 235 – unnecessary comma after isolates

Delete

Line 236 – should read ‘At the present time the function …..’

Done

Line 240 – unnecessary comma after cases

Done

Line 240 – should read ‘suggesting that the influence….’

Done

Line 252 – unnecessary comma after indicates

Done

Line 262 – the sentence should start with ‘The’

Done

Line 269 – change close to closely

Done

Line 276 – full stop in the wrong place

Done

Reviewer 2 Report

Th MS entitle "Genetically related Mycobacterium bovis strains displayed differential intracellular growth in bovine macrophage" describes a relation between the bacterial lineage identified by spoligotyping and virulence (intracellular growth of bacteria and phagocytosis). 

The result is interesting as it contradict to the author hypothesis and to other studies using closely related species, M. tuberculosis, which showed that different geographically predominant M. tuberculosis isolates exhibit different virulence and pathogenicity. Although ita hs interesting result, the presentation needs an English proofread prior to publication. Therefore, I recommend this MS to be published in Veterinary science with some modification. 

Overall, the study is simple yet interesting and provide important knowledge for the field. 

The authors conclude that bacterial virulence of M. bovis is independent from the bacterial lineage identified by spoligotyping. I agree with that, however, the author should address that we should not only rely on currently available genotyping method. Because commonly used genotyping method for M. bovis is currently very limited and still need further improvement. Further analysis, such as using Next-generation sequencing, is necessary to provide detail knowledge supporting this study. If there is any reports describing whole genome analysis of the bacteria, it may improve the discussion part.  

Line 4: complex strains? I dont think "complex" is appropriately used in this sentence. I recommend it to be deleted. 

Line 55-66: this paragraph is confusing and doesn't connect to the previous paragraph. Please modify. 

Line 244 : it is a repeated explanation similar to the one in the introduction line 55 - 66. Please modify. 

Line248: please change "our data" to "our previous study".  

Author Response

Response to reviewers (in red) point by point. Please see the attachment with the manuscript.

Reviewer 2

The authors conclude that bacterial virulence of M. bovis is independent from the bacterial lineage identified by spoligotyping. I agree with that, however, the author should address that we should not only rely on currently available genotyping method. Because commonly used genotyping method for M. bovis is currently very limited and still need further improvement. Further analysis, such as using Next-generation sequencing, is necessary to provide detail knowledge supporting this study. If there is any reports describing whole genome analysis of the bacteria, it may improve the discussion part.  

Done

Line 48: complex strains? I dont think "complex" is appropriately used in this sentence. I recommend it to be deleted. 

Done

Line 55-66: this paragraph is confusing and doesn't connect to the previous paragraph. Please modify. 

Done

Line 244 : it is a repeated explanation similar to the one in the introduction line 55 - 66. Please modify. 

Done

Line248: please change "our data" to "our previous study".  

Done
